# Spectroscopic Techniques and Hydrogen-Sensitive Compounds: A New Horizon in Hydrogen Detection

**DOI:** 10.3390/s24103146

**Published:** 2024-05-15

**Authors:** Bu Si, Yan Hu, Longchao Yao, Qiwen Jin, Chenghang Zheng, Yingchun Wu, Xuecheng Wu, Xiang Gao

**Affiliations:** 1State Key Laboratory of Clean Energy Utilization, Zhejiang University, Hangzhou 310027, China; 12027099@zju.edu.cn (B.S.); yaolc@zju.edu.cn (L.Y.); zhengch2003@zju.edu.cn (C.Z.); wuyingchun@zju.edu.cn (Y.W.); xgao1@zju.edu.cn (X.G.); 2Ningbo Innovation Center, Zhejiang University, Ningbo 315100, China; nb21042@zju.edu.cn

**Keywords:** hydrogen sensor, optical fiber, sensitive materials

## Abstract

Detecting hydrogen leaks remains a pivotal challenge demanding robust solutions. Among diverse detection techniques, the fiber-optic method distinguishes itself through unique benefits, such as its distributed measurement properties. The adoption of hydrogen-sensitive materials coated on fibers has gained significant traction in research circles, credited to its operational simplicity and exceptional adaptability across varied conditions. This manuscript offers an exhaustive investigation into hydrogen-sensitive materials and their incorporation into fiber-optic hydrogen sensors. The research profoundly analyzes the sensor architectures, performance indicators, and the spectrum of sensing materials. A detailed understanding of these sensors’ potentials and constraints emerges through rigorous examination, juxtaposition, and holistic discourse. Furthermore, this analysis judiciously assesses the inherent challenges tied to these systems, simultaneously highlighting potential pathways for future innovation. By spotlighting the hurdles and opportunities, this paper furnishes a view on hydrogen sensing technology, particularly related to optical fiber-based applications.

## 1. Introduction

Hydrogen, with its renewable and potent energy characteristics, presents an appealing option for clean energy applications. Extensive research has been conducted on its energy generation, transportation, and storage. However, hydrogen’s safety profile raises concerns, given its potential for explosion ranging from 4% to 75% in atmospheric conditions [1]. This safety aspect has restrained its broader adoption in the industrial and transportation sectors. Furthermore, vigilant monitoring of hydrogen leakage assumes importance because of its propensity to escape due to its small dimensions and low density, leading to hydrogen embrittlement. A method for testing hydrogen gas leaks at a distance is required to ensure the safety of hydrogen use and transportation. Consequently, optical fiber has emerged as a safer alternative for hydrogen detection based on its extensive use in temperature monitoring, mechanical engineering, and construction owing to its intrinsic physical and chemical attributes. Diverse types of optical hydrogen sensors have been developed, encompassing evanescent optical sensors, surface plasmon resonance (SPR) optical sensors, micromirror sensors, and fiber Bragg grating (FBG) sensors [2]. A synergistic integration of these sensor variants holds the potential to enhance their collective detection capabilities.

The traditional optical fiber sensor is an intricate assembly of several components: a light source, an optical spectrum analyzer, the actual optical fiber, and a coupler. By attaching a gas-sensitive material to the fiber, reactions with the gas can induce optical material property change, subsequently altering light properties, such as wavelength or intensity. By scrutinizing these variations, the concentration of H_2_ can be measured. This method is simple yet efficient, boasting a robust resistance to interference. It enables detection over considerable distances and is both safe and reliable.

To detect H_2_ concentration, two independent spectral anchoring mechanisms are employed. There exist two primary reaction mechanisms upon which mainstream fiber-optic hydrogen sensors are predominantly based. The first mechanism determines hydrogen concentration by observing wavelength shifts, such as changes in the wavelength corresponding to the center reflection signal of FBG (refer to Figure 1) when hydrogen is present. Typical sensors utilizing this mechanism include fiber Bragg grating sensors and interferometric sensors. The second mechanism determines hydrogen concentration by monitoring changes in light intensity, for example, variations in light intensity reflected by fiber micromirrors when hydrogen is present. Representative sensors employing this method include fiber micromirror sensors, fiber plasmonic resonance sensors, and fiber evanescent field sensors.

Direct gas testing with fibers or gratings can assess gases like acetylene based on gas absorption spectroscopy [4]. However, the absorption spectrum of hydrogen is notably weaker; thus, some hydrogen-sensitive materials are required to aid in testing. Materials sensitive to hydrogen, such as noble metals and metal oxides, must be attached to fibers for hydrogen detection. These materials undergo structural changes when reacting with H_2_ and primarily undergo two distinct typical reactions, with the first being reversible under anoxic conditions, suitable for concentration and leak detection. The equations describing the relevant chemical reactions are as follows [5]:(1)$$\left(\alpha \right)Pd+\frac{k}{2}H_{2}↔\left(\beta \right)PdH_{k}$$

If just detecting gas leakage, the hydrogen-sensitive material acts as a catalyst when it appears in the atmosphere. The reaction of metal-loaded metal oxides with hydrogen gas is another principle [6], and the chemical reaction equation is described as follows:(2)$$WO_{3}+xH_{2}\overset{Pt}{\to }WO_{3-x}+xH_{2}O$$
(3)$$WO_{3-x}+\frac{1}{2}O_{2}\overset{Pt}{\to }WO_{3}$$

This article primarily analyzes optical hydrogen sensors based on hydrogen-sensitive materials from the perspectives of materials and processes. By examining their characteristics, we discern this technology’s prospects and challenges, aiming to enhance its application effectively.

## 2. Materials

### 2.1. Metal Oxide (Metal Attached)

Metal oxide with metal attached is widely used in hydrogen detection, such as Pt-WO_3_ [7]. Its performance can be further improved via structural optimization and metal doping. For example, Yang et al. developed a typical material, mesoporous tungsten oxide nanoflower. The response exceeded 700 pm (center wavelength offset) per 1% of hydrogen and had good repeatability [7]. Yan reported nanocrystalline tin oxide thin films for hydrogen sensing based on an optical fiber evanescent field [8]. Pt-WO_3_ is particularly sensitive in gas detection, with a sensitivity of over 1 nm of 1% H_2_ increase and an excellent response time of less than 1 min in the experiment [9]. Faster detection time ensures safety. While these materials have many advantages, some things could be improved and considered. One of the most obvious is their reusability, as they are not easy to reuse for high-concentration tests based on the mechanism of hydrogen reaction with oxygen. Despite this, various microstructures can be created using these metal oxides to improve their responsiveness and immunity to interference. Moreover, one metal oxide (e.g., tungsten oxide) can be paired with several noble metals, such as the hydrogen-sensitive film based on WO_3_-Pd_2_Pt-Pt synthesized by Wang in 2019 [10].

### 2.2. Metal

Metal-based hydrogen sensors differ from metal–metal oxide-based hydrogen sensors. Their principle lies in the reversible reaction between the metal (such as palladium) and hydrogen. These sensors offer a wider concentration range for hydrogen measurement [2]. The metal-based film also outperforms metallic oxide on specific parameters, including response time, which can be less than metal oxide [11]. Composite metal materials or alloy films, such as PdAu alloy nanocones, are also applied to improve performance and meet specific requirements [10]. Furthermore, the sensitivity of metal materials is determined by their thickness, and a suitable thickness ensures the system’s safe and effective operation. Due to the customizability of metallic elements, there is promising potential for their enhanced development in the sensor domain in the future.

### 2.3. Other Materials and Structures

An increasing number of new materials, such as graphene and MOF, are being developed for hydrogen sensors. These materials can offer improved mechanical properties or higher specific surface areas for gas diffusion and can be combined with traditional materials to create multilayer structures for better sensing properties.

Graphene has a substantial specific surface area, superior electrical and thermal conductivity, and outstanding mechanical properties [12]. Graphene or graphene oxide can serve as an excellent substrate, facilitating better integration with metals such as Pd, thus maximizing the metal’s ability to bind with hydrogen. Alkhabet developed an optical sensor based on graphene oxide and metal [13]. By using Pd–graphene oxide (Pd–GO) nanocomposite as a sensing layer, the research taps into the potential benefits of these advanced materials for gas detection. The sensitivities and relatively swift response and recovery times achieved underscore the sensor’s practical utility.

MOFs (metal–organic frameworks) are porous materials of metal ions or clusters connected by organic ligands. MOFs have a high surface area and tunable pore sizes, which makes them useful in various applications such as gas storage, separation, and catalysis. Researchers have also explored the potential of MOFs as hydrogen sensors in recent years due to their high sensitivity and selectivity towards hydrogen gas. Chen developed an optical fiber hydrogen sensor based on MOFs, which employs an optical fiber Mach Zehnder interferometer (MZI) [14]. UiO-66-NH_2_ (MOFs) is used as a hydrogen-sensitive film, and its reaction results in corresponding changes in the effective refractive index and light intensity. These changes are due to the dielectric constant being altered by the reaction.

The material’s structure can also influence the gas detection capability, and in some cases, the structure of specific materials can be tailored to optimize their hydrogen sensing properties. Multilayer materials can be used for FBG, SPR, or fiber-evanescent field sensors. Yan has developed a micromirror-based fiber-optic hydrogen sensor with multiple layers of hydrogen-sensitive material at the end of the fiber [7]. Three layers of hydrogen-sensitive materials are Pd, WO_3_, and Au from the outside to the inside. The three materials are closely arranged and loaded by the sputtering plating method. Tungsten oxide layers are synthesized due to oxygen during the tungsten spray-coating process, and the surface of Pd is engraved with grating. Based on the layered material, the sensor is sensitive to hydrogen detection, has fast response, and has low-temperature characteristics. A wavelength shift of 28 nm is achieved as hydrogen concentration varies from 0 to 4%.

Some typical microstructures of hydrogen sensors were assessed in a fiber-optical system, which might be able to enhance the SSA (specific surface area). Du et al. synthesized material with nanoflower structures based on an in-fiber Mach–Zehnder interferometer for hydrogen sensing [15]. This material is susceptible, with a sensitivity of more than 1000 pm/% (vol%), and possesses a greater specific surface area for gas contacts. The Pt-loaded WO_3_ nanoflowers, when paired with an advanced hydrogen measurement system (Mach–Zehnder interferometer (MZI) high-sensitivity fiber-optic hydrogen sensor (with sensing material embedded within the optical fiber), can better demonstrate their advantages. An electron microscope image of the nanoflowers is shown in Figure 2.

## 3. Technical Analysis

### 3.1. Fiber Bragg Grating (FBG)

In traditional fiber Bragg grating (FBG) sensors, hydrogen concentration is calibrated by monitoring the central wavelength of reflected light (only a narrow band of light around the center wavelength is reflected) (Figure 1 and Figure 3). The traditional fiber Bragg grating (FBG) sensing system is based on central wavelength shift monitoring (Figure 3). This shift might be more significant when the concentration of hydrogen increases. A schematic diagram of wavelength shift is shown in Figure 3. FBG hydrogen measurement was first seen at the end of the last century. FBG is at the core of dimming and hydrogen reactions, so the manufacturing process of FBG is also critical. Currently, there are primarily two methods for fabricating fiber Bragg gratings (FBGs), namely, the ultraviolet (UV) phase mask technique and femtosecond laser inscription [16]. Traditional FBGs manufactured using the UV laser phase mask method require the sensitization of the optical fiber and are unsuitable for high-temperature (around 400 °C) applications [17]. In contrast, femtosecond lasers can inscribe various FBGs on almost all optical fibers. Moreover, FBGs induced by femtosecond lasers exhibit superior thermal stability, making them well suited for sensing in harsh environments.

Sutapun developed a pioneering fiber Bragg grating-based hydrogen sensor in 1999, serving as a new optical hydrogen detector type [18]. The sensor employs a Pd film, a hydrogen-sensitive material, applied to the fiber’s surface. The hydrogen content is detected through deformation caused by hydrogen absorption into the Pd film. This sensor can monitor and provide alerts for hydrogen gas concentrations in the 0.3–1.8% range. This sensor formed the prototype of the current traditional FBG hydrogen measurement, and most subsequent FBG hydrogen measurement sensors have been based on this design.

Caucheteur developed a hydrogen-sensitive material based on metal oxides, and this type of hydrogen-sensitive material and hydrogen reaction requires the participation of precious metals (Pd, Pt, Ag, etc.) and metal materials as reactant in the reaction [19]. Because of its unique wavelength regulation mechanism, FBG can be used with other fiber-optic sensors to achieve a better sensing effect. In addition, the series of different FBGs can also reduce the influence of temperature on the sensor.

Ma developed an FBG-based hydrogen detection device with a thick Pd film as the hydrogen-sensitive material and added polyimide to improve stability to detect hydrogen in transformer oil [20]. The wavelength shift of the FBG sensor is proportional to the hydrogen concentration in transformer oil, and this sensitivity is not affected by temperature. This sensor effectively reduces interference and offers the advantages of fast detection, localization, and real-time monitoring.

In some cases, FBG sensors can also be designed to measure changes in light intensity, such as in the tilted FBG hydrogen sensor developed by Zhang [20]. This sensor uses a nanocomposite film made of Pd and Au as the sensing material, which reacts to changes in hydrogen concentration by altering the intensity of transmitted light. The sensor’s output is then measured using a spectrometer, and a hydrogen concentration–light intensity curve is plotted based on the changes observed. According to data analysis, the sensor showed a high sensitivity of 1.597 dB% (the light intensity in the transmission spectra increased from −11.3068 dB to −10.1788 dB as the hydrogen concentration rose from 0.00% to 1.02%), with an average response time of 37 s and an average recovery time of 49 s.

### 3.2. Interferometric Hydrogen Measurement

Most fiber-optic hydrogen sensors utilize the principle of light interference within the fiber using equipment such as the Mach–Zehnder and Fabry–Perot interferometers, among others [21]. These sensors are based on optical path differences (Figure 4) and have two parallel optical paths, one coated with a hydrogen-sensitive material and the other uncoated. When hydrogen reacts with the hydrogen-sensitive material, it causes a difference in the optical path. This difference leads to light interference at the end of the fiber, resulting in a spectrogram with distinct peaks or valleys of light intensity at specific wavelengths (central wavelength). As the concentration of hydrogen changes, the central wavelength shifts, like hydrogen testing based on fiber Bragg gratings, with the concentration of hydrogen corresponding to the wavelength shift (Δλ). Generally, interferometric sensors are divided into Fabry–Perot and Mach–Zehnder (MZI) interference, depending on the response mechanism. This interference mode may be used with fiber Bragg gratings to exploit both mechanisms’ advantages fully.

#### 3.2.1. Mach–Zehnder Interferometer (MZI)

In 1984, Butler introduced a hydrogen sensor using a Pd film on an optical fiber for hydrogen testing [22]. This sensor was based on the Mach–Zehnder interference principle, one of the earliest fiber-optic methods for hydrogen detection.

Nowadays, MZI sensors show a high sensitivity and a wide dynamic range. The sensor can be prepared using a single optical fiber instead of the traditional double optical fiber. Gu et al. developed a hydrogen sensor based on MZI using PdAu nanowires as the hydrogen-sensitive material [23]. To address the weak mechanical stability of pure Pd, this study utilized the PdAu nanowires to enhance the strength of the hydrogen-sensitive material. The hydrogen-sensitive material was loaded onto the test fiber while the reference fiber remained unloaded, creating an optical range difference to produce an interference spectrum under hydrogen intervention. The troughs of the spectra shifted at different hydrogen concentrations, allowing for the calibration of hydrogen concentrations. The device exhibits reversibility and low energy consumption. The principal diagram of this H_2_ sensor is shown in (Figure 5).

MZI interference can also be achieved with a single fiber by changing the structure of the fiber. Du developed a susceptible single fiber-based MZI hydrogen sensor using a graded-index fiber (GIF) to test for hydrogen [15]. A cavity filled with polymer is formed by intervening with a laser, creating two optical paths—one each for the GIF and the hole. The entire region is wrapped with hydrogen-sensitive material (Pt-loaded WO_3_), and the two internal rays can interfere with each other when hydrogen is involved. This MZI-based hydrogen sensor boasts a sensitivity of 1948.68 nm/% (vol% hydrogen) at hydrogen concentrations close to 0.8%. The sensor is highly responsive and can detect low concentrations of hydrogen, making it a valuable tool for gas-sensing applications.

#### 3.2.2. Fabry–Perot Interferometer (FPI)

In 1994, Zeakes developed a hydrogen sensing system based on the Fabry–Perot interferometer (FPI) [24]. A sputter-coated Pd film serves as the hydrogen-sensitive material for the optical fibers. This configuration utilizes two optical paths (one affected by the hydrogen-sensitive material) in the same fiber, leading to interference. The visual range of one of the paths changes when hydrogen is present. An extrinsic Fabry–Perot interferometric sensor was used for hydrogen testing, and its configuration effectively avoids polarization issues and monitors a single axial strain. This interference-based method of measuring hydrogen is more sensitive, and the lower detection limit is predicted to reach 32 ppm.

Fiber-optic hydrogen measurement systems with multiple micromirrors in series are an advanced technology that typically incorporates light interference. A multimicromirror hydrogen measurement system was developed [25] with a fiber tail consisting of a section of large-mode-area fiber (LMAF) and a section of hollow core fiber (HCF) (Figure 6), with the end coated with a hydrogen-sensitive material. The hydrogen-sensitive material on the back is Pt-doped WO_3_/SiO_2_ powder for hydrogen reaction. This system achieved a sensitivity of 1.04 nm/% for hydrogen in the range of 0.3–2.4%, with a response time of only 80 s.

Lee proposed a double-C structure in 2018 that exhibits excellent performance as a combination of micromirrors and Fabry interference [26]. The most striking feature is its structural design, which consists of a cavity at one end of the fiber (double-C structure), with the inner C cavity forming a microcavity Fabry–Perot interferometer (FPI) and the outer C cavity with an embedded Pt-loaded WO_3_/SiO_2_ powder as the hydrogen-sensitive region. There are some complicated metallic WO_3_/SiO_2_ systems with Pt loaded as H_2_ sensor, with only 30 s response time and 17.5 nm/% H_2_ sensitivity that have shown good performance in H_2_ sensing. Moreover, an FBG can be combined with this structure to test the temperature and achieve versatility.

### 3.3. MultiFBG Test System

A dual-FBG system can effectively harness the modulation capability of FBGs. By employing two FBGs with different central wavelengths in a hydrogen sensing setup, often coupled with fiber micro-lenses, the relative intensities of light from these FBGs can accurately indicate the concentration of hydrogen gas [9].

Dai developed a micromirror hydrogen sensor that employs two sets of Bragg gratings on an optical fiber, with one set of high-reflectivity gratings and another set of low-reflectivity gratings [27]. The hydrogen-sensitive films attached to the fiber’s end comprise three layers: 10 nm Pt, 20 nm Pd_2_Pt, and 200 nm tungsten oxide. Hydrogen does not significantly affect the high-reflectivity gratings, whereas the low reflectivity gratings exhibit a more noticeable change in reflectivity. The light intensity ratio (I_1_/I_2_) of the two FBGs provides effective calibration of the hydrogen concentration and minimizes interference from humidity. Moreover, this sensor is sensitive and can monitor gases down to 10 ppm.

These techniques can be upgraded for additional testing environments, such as adding heating equipment. Wang developed a fiber-optic hydrogen sensor based on a dual FBG with a hydrogen-sensitive film and a heating device [10]. The unique heating system (PID algorithm to control the output power) can ensure uniform and controllable temperature, with the hydrogen-sensitive material comprising 150 nm WO_3_, 40 nm Pd_2_Pt, and 5 nm Pt. The increase in I_1_/I_2_ indicates an increase in hydrogen concentration, which can be calibrated as a function of the data. Furthermore, the heating system can reduce the influence of ambient temperature fluctuations, leading to the stability of the hydrogen sensor.

### 3.4. FBG with FPI

The FBG (fiber Bragg grating) exhibits a high reflectivity for specific wavelengths, acting somewhat like a selective mirror for certain wavelengths. Therefore, any variations in the intensity of the reflected light or the utilization of FBG for optical interference are particularly pronounced [10]. This has led to the development of hydrogen sensors that combine both FBG and FPI (Fabry–Perot interferometer).

Luo’s fiber-optic hydrogen sensor is a highly advanced design incorporating a fiber Bragg grating with a micromirror to create a superior hydrogen sensor [5]. The Bragg grating is inscribed in the middle of the fiber, while the end of the fiber is coated with a hydrogen-sensitive material, specifically Au–Pd–graphene. To enhance the sensor’s performance, it uses a short hollow-cavity length and a slit end on the hollow-core fiber to reduce dispersion loss and insertion loss effectively. In addition, the end of the hollow-core fiber is loaded with a multilayer hydrogen-sensitive material, making it even more sensitive. This design produces a central wavelength shift of 290 pm (near the central wavelength of 1550 nm) when the hydrogen concentration is 1.5–4.5%, indicating a susceptible hydrogen sensor that can detect low levels of hydrogen in each environment [5].

### 3.5. Mirror H_2_ Sensor

A mirror fiber-optic hydrogen sensor effectively measures hydrogen concentration through a change in intensity at a typical wavelength [28]. Unlike a fiber Bragg grating, the hydrogen-sensitive material film is located at the end of the fiber. As the film reacts with hydrogen, it undergoes changes that effectively alter the refractive index of the system, leading to a shift in the intensity of the reflected light. The change in reflectivity allows for calibration of the hydrogen concentration. The film’s selection and attachment method are crucial to achieving a response to hydrogen, and the strength and toughness of these materials must be considered during design. A schematic diagram of the mirror H_2_ sensor is shown in Figure 7. It is important to note that the sensor’s performance is highly dependent on the hydrogen-sensitive film’s quality and consistency and the attachment method’s stability.

Butler developed a micromirror-based fiber-optic sensor in 1993 to test higher hydrogen gas concentrations [29]. The sensor utilized a 10 nm-thin Pd film as the hydrogen-sensitive material and tested hydrogen concentrations of up to 10% in nitrogen.

In 2000, Bevenot developed a fiber-optic sensor to detect hydrogen leaks in the cryogenic engines of European rockets [30]. This sensor used a commercial multimode fiber-HCN H type (diameter 400 μm) with the sensing area coated on the bottom of the fiber with a Pd film. The Pd film can be efficiently heated using high-power laser heating, enabling hydrogen testing in the temperature range of −196 °C to +23 °C. The sensor has a hydrogen detection range of 1% to 17% and a response time of less than 5 s.

However, this sensor was susceptible to interference from intensity fluctuations in the optical sensing system, so Dai reported a hydrogen sensor based on a micromirror sensor with improvements [31]. The hydrogen-sensitive material used in this sensor was WO_3_–Pd_2_Pt–Pt nanocomposite film (Figure 8). The reflected mechanism was based on the ratio of light intensity (I_1_/I_2_) to calibrate the hydrogen concentration, with reference intensity (I_1_) and sensing intensity (I_2_), where the fluctuation of I_1_ and I_2_ was also lower. The sensor had a fast response time of around 20 s when the hydrogen concentration was 1000 ppm, and at that time, the response value I_1_/I_2_ was 0.38 (0.45 when there was no hydrogen). The sensor had good resolution at 100–5000 ppm and showed promise for future use.

### 3.6. Surface Plasmon Resonance (SPR) Sensor

An optical fiber surface plasmon resonance sensor is also a standard hydrogen sensor based on changing light flux to sense hydrogen concentration [32]. Fiber-optic plasma resonance refers to the resonance between the vanishing wave and the plasma wave under the intervention of the outside world, resulting in the absorption of the energy of the incident light; that is, the intensity of the transmitted light becomes weak (Figure 9). This technology was first applied to gas sensing in 1982, laying the foundation for future developments [33].

Benson’s work in 1999 successfully detected hydrogen at an explosive concentration of 4% using the SPR process [34]. Tungsten oxide was used as the hydrogen-sensitive material for optical fibers, and Pd was used as a catalyst for WO_3_ to accelerate the reaction. The decay in reflected energy was used to calibrate the hydrogen concentration. This study succeeded in fiber-optics, chemical color change, and hydrogen detection. However, this sensor may produce erroneous results when water vapor is present.

Ohodnicki developed an SPR-based fiber-optic hydrogen sensor that can detect hydrogen concentrations between 1% and 100%, giving it the advantage of testing high concentrations of hydrogen [35]. This principle can be used to test hydrogen at higher concentrations, up to 20% in the infrared band, above the detection limit of traditional FBG.

In addition, the region of surface plasmon resonance can be located at the tail end of the fiber, and the light intensity can be used to detect hydrogen. Kim proposed a surface plasmon resonance hydrogen sensor in which the hydrogen-sensitive material is Pd-plated gold nanoparticles [36]. The composite sensitive film is deposited on the tail end of the fiber using a layer assembly method, where the gold particles are coated with a Pd layer on the surface. The sensor has a response time of 116 s in the 0.8–4% range, and the lower detection limit can be as low as 0.086%. The sensor is also reusable.

### 3.7. Optical Fiber Evanescent

An optical fiber evanescent field-based hydrogen sensor is a reliable method for detecting hydrogen concentration based on changes in light intensity. This sensing method relies on the fact that some light energy is lost through the fading field, which can be used to control the intensity of the projected light (transmittance). At the same time, this sensor is like the SPR sensor (Figure 10), but the reaction mechanism is slightly different. In an SPR sensor, the light transmission is weakened due to resonance, whereas in the evanescent field-based sensor, the diluted energy loss is due to the evanescent field.

Tabib-Azar (1998) successfully operated a fiber-connected MEMS pressure sensor using a fiber-evanescent field in the 0–20 psi pressure range [37]. In the same year, they also developed a gas sensor based on the evanescent field of optical fibers [38]. It could detect hydrogen concentrations between 0.2% and 0.6% with a 100 nm-thick Pd layer and a 20- to 30-s response time. Low temperatures did not affect the sensor response, and the response time was only doubled at −10 °C.

However, there is still room for improvement in hydrogen measurement using the evanescent field of optical fibers, and the use of D-type optical fibers may help address this issue. Cao reported a temporary field hydrogen sensor based on D-type optical fibers with Pd alloy as the hydrogen-sensitive material [11]. In this sensor, the change in luminous flux (light intensity) due to hydrogen reacting with the hydrogen-sensitive material is recorded as the change in I/I_0_. I represents the light intensity with hydrogen, and I_0_ represents the intensity without hydrogen. The addition of the alloy effectively reduces the influence of PdH_x_ phase transformation and improves the response speed. Moreover, adding nanostructures further enhances the response effect. This sensor can detect hydrogen concentrations from 0.25% to 10% at atmospheric pressure with a response time of 30 s at a 4% volume concentration of hydrogen.

### 3.8. Sensor Structure

Of course, many other structures are used for optical hydrogen gas monitoring. While these structures operate on principles like fiber-optic hydrogen sensors, their unique designs might enhance performance.

#### 3.8.1. Structure of Fiber Sensor

In gas detection, the specific surface area is a crucial parameter that benefits the gas reaction. In addition to the porous structure of the sensing material, the unique structure of the optical fiber can also be used to improve the contact area. For instance, Zhou developed a single spiral and a double spiral on the cladding of the FBG area via laser engraving [39]. The hydrogen-sensitive material can be attached to the fiber’s surface. Some of it goes into the groove, effectively increasing the contact area and making it easier for the hydrogen-sensitive material to react with hydrogen. This device achieved an extensive test range and fast response time. Humidity resistance was also improved, effectively improving hydrogen-testing capabilities, as shown in Figure 11.

In addition to laser engraving, pre-polishing the fiber and grinding micro-grooves along the direction of the fiber on the cladding can also improve performance. Due to grooves, there are multiple symmetrical inward gaps in the cutting surface of the optical fiber, and these gaps can be coated with sensing materials. Karanja took advantage of this structure and used silver as a hydrogen-sensitive material, resulting in a system with good repeatability [40]. Zhou developed a new spiral structure of optical fiber combined with various nanomaterials, which reacted quickly within several seconds with 0.02% and 4% hydrogen flowing into the gas chamber [41].

Hollow fiber is another fiber that can be used for hydrogen testing. Liu developed a new type of fiber with hydrogen-sensitive material inside a hollow fiber, calibrated according to the central wavelength shift [42]. EVA–Pd coating HF is fabricated by sequentially depositing Pd and EVA films on the inner surface of quartz capillaries while controlling the solution’s deposition time, flow rate, and temperature to prepare a uniform coating. This configuration resulted in a protected sensing film and an average sensitivity of 2.66 nm/% over the 0–4% hydrogen concentration range.

Some structures can be synthesized and doped with hydrogen-sensitive materials during synthesis. Guo developed a synthetic composite porous optical fiber of silica-doped Pd synthesized by the sol–gel method [41]. The integrated synthesis process results in a considerable improvement in sensitivity, with the sensor being reversible for hydrogen testing and capable of monitoring hydrogen at 1% concentration.

#### 3.8.2. Cantilever Sensor

Hydrogen sensors based on cantilever structures have gained considerable attention in recent years. One such structure is the catch-and-release cantilever structure, which is like the micromirror structure. In 2020, Xiong designed a combined fiber-optic and mechanical cantilever hydrogen measurement system, which utilizes a micro-cantilever loaded with a Pd film for hydrogen monitoring [43]. The hydrogen-sensitive micro-cantilever was synthesized using femtosecond laser-induced two-photon polymerization (TPP), and the surface was coated with a Pd film using magnetron sputtering, as shown in Figure 12. The reaction of the Pd film on the microcantilever with hydrogen produces a deformation that drives the entire microcantilever to bend, thus changing the central wavelength of the reflected spectrum of the system. This sensor exhibits a strong response, with a sensitivity of 2 nm/% hydrogen, as the hydrogen concentration ranges from 0 to 4.5%, and it has a response time of about 13.5 s at 4% (*v*/*v*). The hydrogen-sensitive micro-cantilever’s innovative design and synthesis process provide a promising platform for developing high-performance hydrogen sensors.

#### 3.8.3. Multiple-Layer Film Sensor

Yi Liu developed a hydrogen gas sensor based on a series of connections of multiple thin hydrogen-sensitive films. Light passes through all the parallel films via a collimator [44]. When used as a material for hydrogen gas, the multilayer thin Pd–Y alloy film can effectively alter the light’s transmittance in the presence of hydrogen gas, as demonstrated by the reaction mechanism. The gas-sensitive films were prepared using physical vapor deposition (PVD) and sputter-coating to deposit hydrogen-sensitive materials onto quartz films. Experimental results indicated that the response time of the hydrogen gas sensor gradually decreases as the hydrogen concentration increases, with the change being nonlinear. The sensor’s shortest response time at a concentration of 4% is approximately 4 s, and the lower detection limit is 0.05%. The schematic diagram and the response curve are shown in Figure 13. Despite their outstanding performance, sensors of this type are cumbersome.

## 4. Sensor Fabrication

Choosing and applying effective hydrogen sensing materials are pivotal for accurate measurements for optical fiber hydrogen sensing systems. Several techniques exist to use these materials in grating, including traditional deposition, spray-coating, chemical vapor deposition (CVD), and metal sputtering. Of these methods, metal sputtering stands out as the favored approach for affixing metal-based materials. Feng pioneered an FBG hydrogen sensor with a sputter-coating technique with Pd–Ni. This method generates free atoms through collisions with external entities like AR ions [45]. To achieve a consistent distribution of the hydrogen-sensitive material across the grating, rotating or spraying the optical fiber from various angles during the coating process is essential.

In addition to metal sputter, other methods, such as sticking metal foil directly onto the optical fiber’s surface, can also load metal materials. Fisser developed a simple method for loading Pd foil as a hydrogen-sensitive material, achieving good performance and excellent sensing performance at a thickness of 20 microns [46]. Experimental data suggested that the Pd foil-type hydrogen-sensitive material sensor performs better than traditional sputter-method sensors of the same thickness. The response wavelength of the Pd foil hydrogen-sensitive material is 0.5 nm compared to the response wavelength of splash-wrapped material of 0.25 nm [46].

Traditional deposition methods can also load hydrogen-sensitive materials onto optical fibers. Ma designed a loading method for hydrogen-sensitive materials using a plating solution of metal hydrogen-sensitive materials as raw material [42]. This method allows for the loading of organic matter or even multilayer hydrogen-sensitive material, such as Pd/EVA hydrogen-sensitive material.

Deposition is an effective way to load hydrogen-sensitive materials framed by metal oxides onto optical fibers in the grating region. Ma developed a micromirror fiber-optic hydrogen sensor using a Pd-decorated multilayer graphene (MLG) hydrogen-sensitive material, where the loading of the MLG method is dipping [47]. The MLG thin layer is specially treated to float on the surface of deionized water, and the tip of the fiber is dipped into the floating layer, forming a hydrogen sensing layer on the tip of the fiber. This method achieved a lower detection limit of about 20 ppm and a short response time of about 18 s using a hybrid film of about 5.6 nm-thick Pd and about 3 nm-thick MLG.

Nugroho pioneered an innovative material-loading technique using a metallic alloy synthesized via high-temperature annealing on a planar substrate [48]. Initially, metallic chromium was layered onto the substrate, complemented by a carbon film. This was succeeded by introducing a Pd–Au alloy nanodisk array via hole–mask colloidal lithography (HCL). Calcination was employed to remove organic materials, after which the chromium was eradicated using an etching solution. This process facilitated the floating of the alloy and carbon film on water. Employing an optical fiber, the carbon film and alloy were captured and adhered to the fiber’s surface. Subsequently, removing the carbon film ensured exclusive alloy adherence to the fiber. The detailed preparation sequence is illustrated in Figure 14.

An effective means for producing synthetic materials, 3D printing will also be widely used in manufacturing optical sensors, leveraging the advantages of additive manufacturing. Iwan Darmadi proposed utilizing 3D-printing techniques (Figure 15) to fabricate hydrogen-sensitive materials [49]. The printing method used is FDM 3D printing (fused deposition modeling), which can integrally produce sensors where polymers and metal elements are organically combined. The sensor not only has a rapid response to hydrogen gas, but its specificity for hydrogen gas has also greatly improved, effectively reducing the influence of other gases such as carbon monoxide. Utilizing various nanocomposite materials for multilayered FDM 3D printing can facilitate the on-site fabrication of customized sensor units with diverse functionalities.

## 5. Specific Environmental Hydrogen Measurement

### 5.1. Temperature Reference

Temperature is also a critical interference factor based on the fiber-optic hydrogen measurement principle (Figure 1). The most popular current method is to connect an empty grating in series to balance the temperature. This grating is called a reference grating. When the test grating and the reference grating are in the same environment, the response value of the test grating minus the response value of the reference grating is the response caused by pure hydrogen.

Based on this principle, Yan synthesized a hydrogen sensor in 2011 that can effectively offset the effect of temperature [50]. Pt–WO_3_ is loaded on the side face of side-polished fiber Bragg grating through magnetic sputtering technology and serves as a hydrogen sensor. Moreover, the relative wavelength is obtained by subtracting the center wavelength of the reference fiber grating from the center wavelength of the sensing fiber grating, which can improve the accuracy of the sensor by reducing the influence of ambient temperature. The sensor recorded peak central wavelength shifts of 25 and 55 pm at 4% and 8% hydrogen concentrations, respectively.

In 2017, Xiang developed a heated FBG hydrogen sensor with improved stability, sensitivity, and response speed during hydrogen exposure [45]. Figure 16 shows a schematic diagram of the heated FBG hydrogen sensor. Introducing a heating system can enhance the sensor’s sensitivity to hydrogen. Overall, these advancements in temperature control and hydrogen sensing have led to more accurate and practical hydrogen measurement systems. A novel hydrogen sensor with inherent temperature compensation and a controllable optical heating system, this sensor holds excellent promise for hydrogen concentration detection in oxygen-free environments.

### 5.2. Special Environment Test

In practical testing scenarios, hydrogen measurement systems may encounter other gases, such as water vapor or hydrocarbons, which can interfere with the accuracy of measurements. However, there are effective methods to mitigate the interference of these gases, such as adding a filter layer or enhancing the material’s performance.

High selectivity is currently a reliable approach, emphasizing specificity towards hydrogen to reduce or eliminate the influence of other gases, especially in mixed-gas environments. Elena developed a fiber-optic hydrogen sensor that operates based on changes in light intensity like surface plasmon resonance (SPR). The sensor demonstrates excellent optical performance and notable specificity [51]. The sensor is founded on a gold-coated multimode optical fiber, leveraging its plasmonic properties, and is adorned with an IRMOF-20 layer known for its high selectivity and affinity to hydrogen. The sensor exhibits an exceptionally high response to hydrogen and boasts robust selectivity. Furthermore, it remains insensitive to humidity. The sensor’s absorption spectrum remains predominantly stable when exposed to gases like carbon dioxide and nitrogen dioxide. Moreover, when hydrogen is combined with these gases, the sensor’s response profile closely mirrors that observed for hydrogen in ambient air. Spectral graphs in the presence of interfering gases are shown in Figure 17.

One approach involves using a zeolite filter membrane in fiber-optic hydrogen measurement systems, significantly enhancing the system’s resistance to interference. Sun developed a multilayer hydrogen-sensitive material involving zeolite [52]. The sensing film comprises Pd–silica (Pd–SiO_2_), while an NaA zeolite membrane serves as a filtering layer to sieve out interfering gases [52] (Figure 18). This zeolite membrane effectively screens out interferences like carbon monoxide (CO) [52]. Results indicate that the zeolite membrane notably mitigates the impact of carbon monoxide through its reactive properties (Figure 18).

This type of sensor uses multiple layers of hydrogen-sensitive film structures made from diverse materials to accommodate complex test environments. Westerwald proposed a multilayer film-based fiber-optic hydrogen sensor, in which the hydrogen sensing region located at the end of the fiber is set up with a multilayer film [53]. The sensor probe is a multilayer structure of polytetrafluoroethylene (PTFE), a Pd–Au sensing layer, and a Ti adhesion layer, which are attached to a multimode fiber. The PTFE film is on the outermost layer and is used to counteract the effects of water vapor, while the titanium layer ensures the mechanical stability of the structure. The sensor is susceptible, with response and recovery times of less than 15 s. The sensor’s response to water vapor (up to 2%) and methane (up to 5%) to hydrogen is unaffected.

Fiber-optic hydrogen sensors can effectively address various issues related to remote hydrogen telemetry in real-world environments. Once these issues are resolved, such sensors might replace existing leak detection tools in areas like hydrogen refueling stations, transformer oils, and nuclear power plants.

## 6. Challenges

### 6.1. Limit of Detection (LOD)

The sensitivity of hydrogen sensors is a critical parameter that determines their effectiveness in ensuring safety. However, a significant challenge facing most hydrogen sensors is their high lower limit of detection (LOD), sometimes near 0.5% [54]. This high LOD makes it challenging to use these sensors in critical areas requiring more sensitive detection capabilities. To address this limitation, researchers have focused on developing new hydrogen sensors with improved sensitivity, especially those that can detect lower hydrogen concentrations with high accuracy [54]. The performance of some optical hydrogen sensors is listed in Table 1.

On the other hand, the upper limit of detection is another significant challenge that needs to be addressed. Oxygen in the reaction mechanism, especially in sensors that use metal oxides as the hydrogen-sensitive material, makes it challenging to test hydrogen concentrations above 4% [1]. In contrast, sensors that use precious metals as hydrogen-sensitive material can detect hydrogen concentrations above 4%, but the concentration–wavelength curve becomes almost horizontal at concentrations above 8% [55]. Furthermore, some optical fiber sensors that rely on light flux to detect hydrogen can detect high concentrations of hydrogen. However, their hydrogen–transmittance curve becomes almost flat at higher hydrogen concentrations, which limits their usefulness in applications that require the detection of very high hydrogen concentrations. The upper limit of detection for fiber-optic hydrogen sensors should also be appropriately increased.

### 6.2. Adsorption and Resolution Time

In critical areas, it is crucial to have hydrogen sensors that can respond rapidly to identify potential risks and enable swift action. Recovery time is also essential, as shorter recovery times can enhance the material’s performance and facilitate reuse. However, balancing response and recovery time remains challenging for many hydrogen sensors, limiting their development. Therefore, it is essential to continue researching and developing hydrogen sensors that can respond quickly and recover rapidly without compromising their accuracy and reliability. Such sensors would significantly improve safety in critical areas by enabling timely identification of risks and quick decision-making.

## 7. Conclusions and Outlook

### 7.1. Conclusions

Over the last few years, several reports of thin film-based optical hydrogen sensors have been reported. Metal-loaded metal oxide and metal materials continue to be the primary choices for hydrogen-sensitive membranes. The metal selection is primarily based on Pd. In summary, the current trend combines various types of hydrogen-sensitive materials. The types of optical hydrogen sensors mainly include surface plasmon resonance (SPR), fiber Bragg grating (FBG), interference, evanescent field, micromirror, and more. The characteristics of these traditional measurement methods have been discussed, and some advanced measurement methods are also mentioned. Meanwhile, these advanced techniques alleviate the impact of factors such as humidity and temperature while expanding the potential applications of hydrogen sensors. Those interested in this field will find the composition of the structural elements of hydrogen-sensitive materials and the usage scenarios of the sensor’s measurement range fascinating.

Optical hydrogen sensors can be essential in some hydrogen testing areas, such as confined spaces. Based on the present trends, future hydrogen sensors will be better equipped to adapt to the changing test environment. It is believed that in the future, the optical hydrogen sensor will partially replace the existing hydrogen sensor and achieve its advantages of high transmission speed, low cost, and anti-strong magnetism. As a result, the sensor will become more versatile, making it easier to detect hydrogen leaks and increase safety in hazardous areas.

### 7.2. Outlook

#### 7.2.1. More Reliable Hydrogen-Sensitive Materials

The mechanical stability of hydrogen-sensitive materials is a critical factor that needs improvement to ensure reliable testing. Deposition or sputtering methods to load the material can result in a thin hydrogen-sensitive layer, which makes it difficult to form a dense material. This can cause cracks that affect stability, recovery time, and material reuse. In some unique environments, adding Ti can improve mechanical strength [53]. However, there is a need for more reliable materials that can guarantee repeatable testing in the future. One possible solution could be to use more ductile materials to reduce the formation of cracks. This will enable reproducible hydrogen testing and more efficient testing in multiple environments.

Polymers are a promising material for optical hydrogen sensors due to their reliability and ability to form cavities in the sensing site. For instance, in one study, a polymer was embedded in a Fabry–Perot hydrogen sensor, which resulted in fast hydrogen response and high response values [58]. The MZI-based hydrogen sensor mentioned above also contains polymer in its interior, which was treated with UV light to enhance its reliability. This sensor demonstrated a similar wavelength shift in several repeated tests with good repeatability. PDMS (polydimethylsiloxane) is a common polymer used in fiber-optic hydrogen sensors to provide protection or performance enhancement. Figure 19 shows a schematic diagram of a PDMS-doped hydrogen sensor. These polymers, such as PDMS, must meet appropriate gas tightness to ensure gas sensing.

#### 7.2.2. Telemetry/Multifunction

Fiber Bragg gratings can perform telemetry due to the excellent filtering properties of Bragg gratings. The principle of the test is that different Bragg gratings are inscribed on a single fiber, and these Bragg gratings have different reflectance or central wavelengths. These differences allow each FBG to be analyzed spectrally independently. This allows the presence of multiple sensors on each fiber, thus forming one or more sensor arrays, with the array arrangement schematic shown below (Figure 20). These arrays based on Bragg gratings have been used to test liquid levels. Barone describes a method for level measurement based on temperature variations (gases and liquids have different temperatures) and using Bragg grating arrays [59]. This type of array may be used for hydrogen testing. At the same time, the hydrogen-sensitive material is coated with different FBGs (with different reflectance or central wavelengths), and the spectra can yield changes in the central wavelength at multiple locations. This configuration would address hydrogen testing at multiple leak points and extend the measurement distance.

There are also some remote sensing methods for hydrogen, such as direct laser testing. These methods have already achieved relatively good results in laboratory experiments. Avetisov developed a laser-based hydrogen sensor that uses wavelength-modulated spectroscopy (WMS) for contact-free molecular hydrogen measurement, which can be used in industrial environments for contact-free and accurate hydrogen measurement [61]. Ma developed a highly sensitive hydrogen sensor based on light-induced thermoelastic spectroscopy (LITES) technology using a continuous wave and a distributed feedback diode laser as the excitation source [62]. The laser emits in the 2.1 μm region and targets the strongest hydrogen absorption line at 4712.90 cm^−1^. To remove noise, a robust shallow neural network (SNN) fitting algorithm is introduced. Furthermore, a heterodyne H_2_-LITES sensor was constructed, achieving a detection limit of 45 ppm.

#### 7.2.3. Concentration Test

Some of the current techniques are rarely used to test high concentrations of hydrogen, and this needs to be improved. Since the lower concentration limit of hydrogen explosion is 4% by volume, hydrogen’s general fiber-optic measurement is selected at a concentration of less than 4% to ensure safety. However, some studies have tested hydrogen above 4%, such as Ohodnicki’s 2015 study of a range of up to 100% hydrogen, but the sensor’s response slows at levels above 10% [35]. In the future, more suitable materials will be used to increase the range.

## Figures and Tables

**Figure 1 sensors-24-03146-f001:**
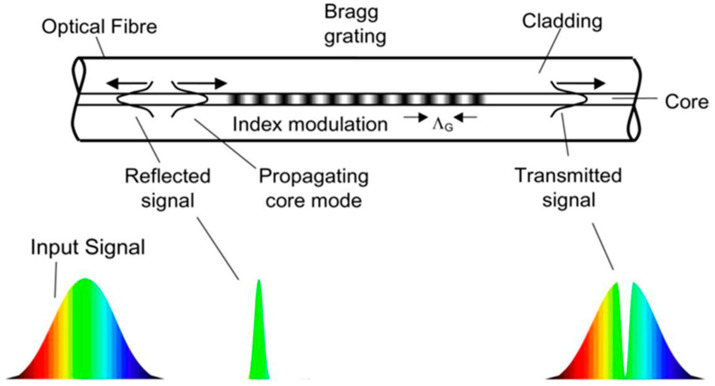
Fiber Bragg grating (FBG) tuning principal diagram (based on central wavelength adjustment) [3]. Note: The arrow indicates the direction of light.

**Figure 2 sensors-24-03146-f002:**
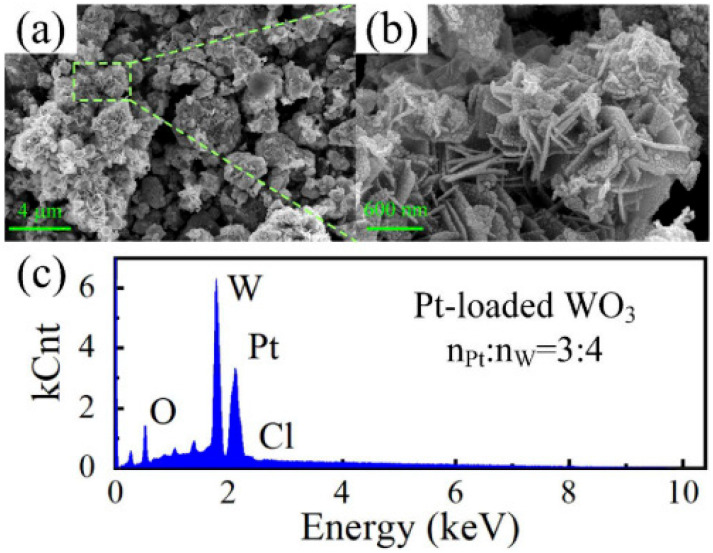
(**a**,**b**) SEM images of Pt-loaded tungsten oxide (WO_3_) nanoflowers at different magnifications; (**c**) EDS image of Pt-loaded tungsten oxide (WO_3_) nanoflowers [15].

**Figure 3 sensors-24-03146-f003:**
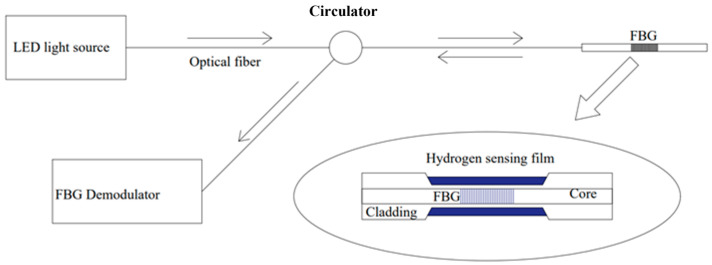
Schematic diagram of traditional FBG sensors. Note: The arrows indicate the direction of light, and blue highlights the distinctive features of this sensor; the big arrow means explanation.

**Figure 4 sensors-24-03146-f004:**
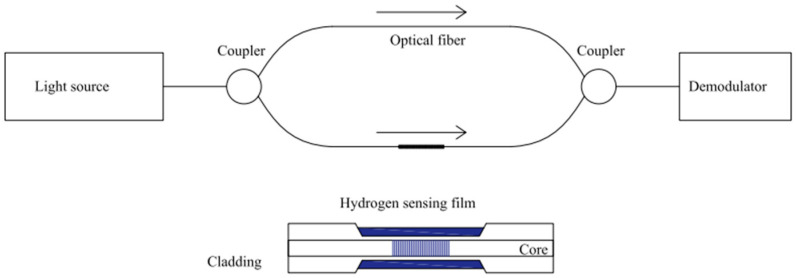
Schematic diagram of optical fiber interference H_2_ sensors. Note: The arrow indicates the direction of light, and blue highlights the distinctive features of this sensor.

**Figure 5 sensors-24-03146-f005:**
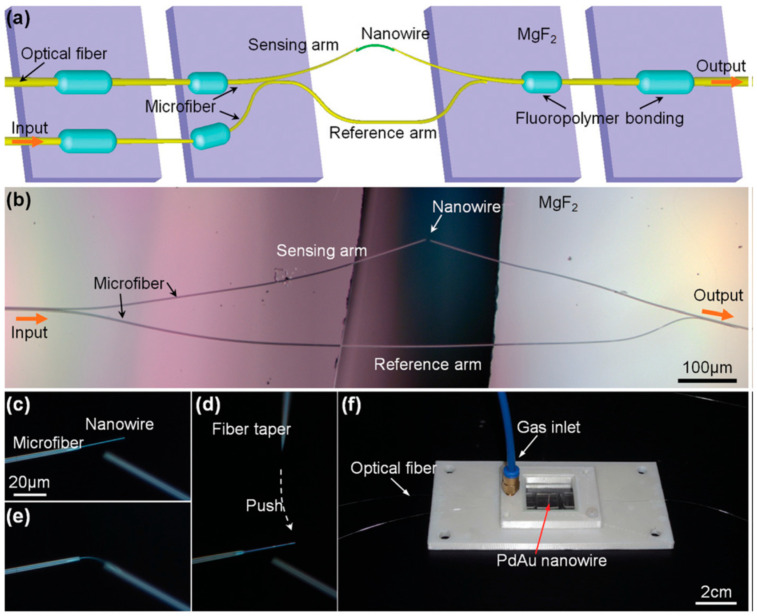
(**a**) Schematic diagram of the MZI hydrogen sensor composite with nanowires and (**b**) microscope image of the sensor. (**c**–**e**) Connections of nanowires to optical fibers using fiber cones. (**f**) Optical micrograph of the MZI sensing device [23].

**Figure 6 sensors-24-03146-f006:**
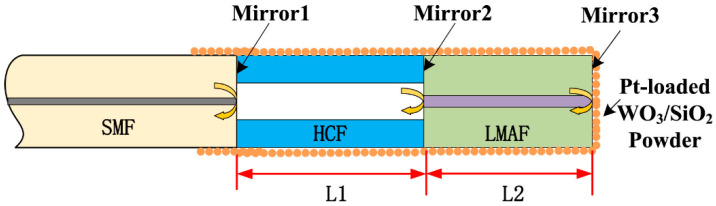
A Fabry–Perot interferometer hydrogen sensor based on multiple micromirrors [25]. The sensing material used is Pt–WO_3_. Note: The yellow arrows indicate the light direction, and the red arrows are length.

**Figure 7 sensors-24-03146-f007:**
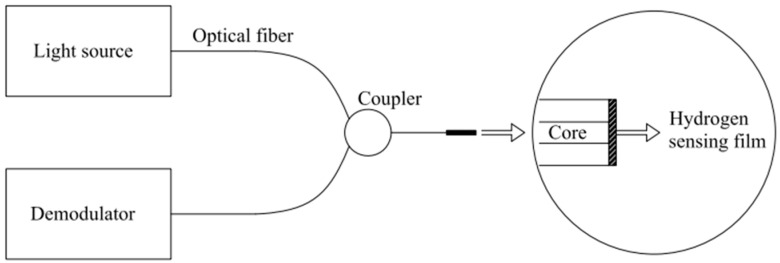
Schematic diagram of a micromirror optical fiber H_2_ sensor. Note: the arrow means explanation.

**Figure 8 sensors-24-03146-f008:**
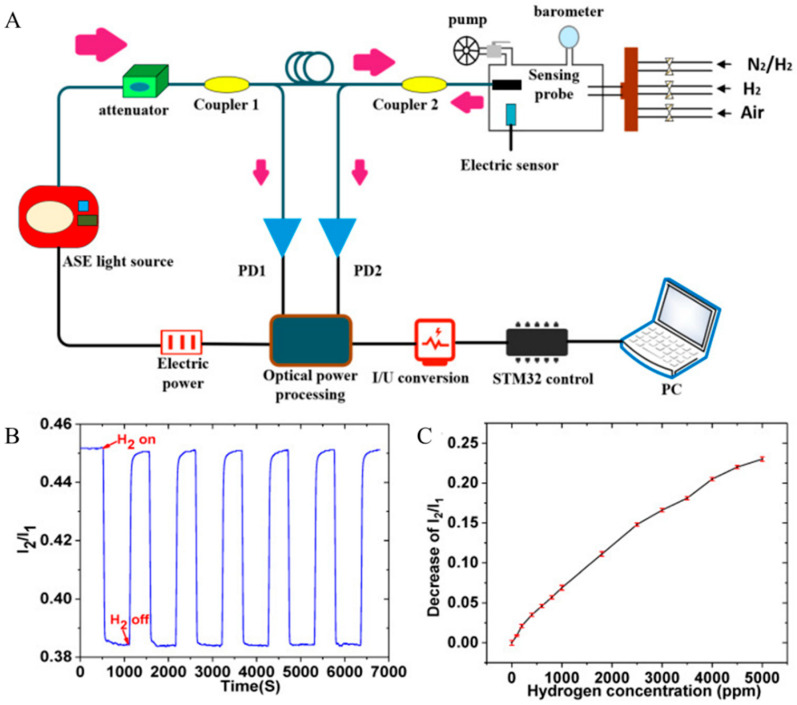
(**A**) Schematic of micromirror H_2_ sensing system. (**B**) Hydrogen response of sensing system with several time cycles. (**C**) Decrease in I_2_/I_1_ under H_2_ concentrations [31]. Note: The arrow indicates the light direction.

**Figure 9 sensors-24-03146-f009:**
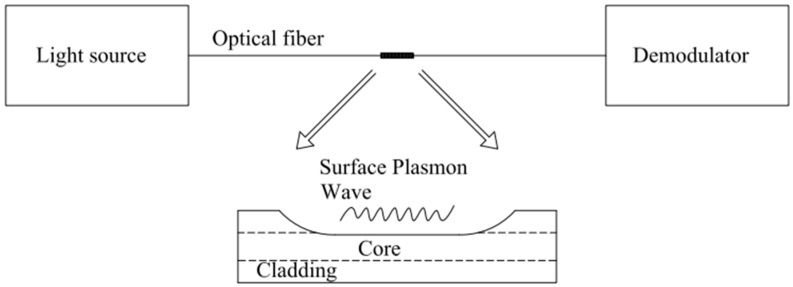
Schematic diagram of SPR H_2_ sensors. Note: the arrow means explanation.

**Figure 10 sensors-24-03146-f010:**
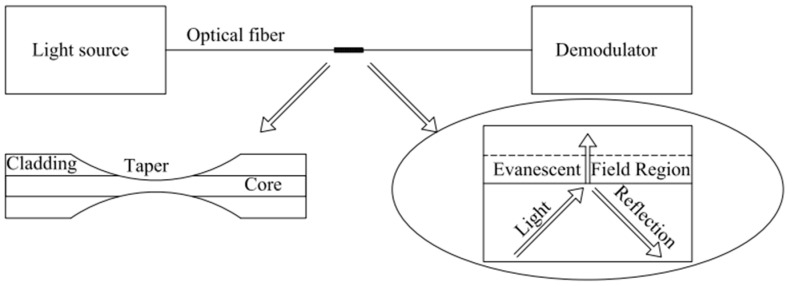
Schematic diagram of optical fiber evanescent H_2_ sensors. Note: the arrows in the main figure mean explanation, and the arrows in the ellipse mean light direction.

**Figure 11 sensors-24-03146-f011:**
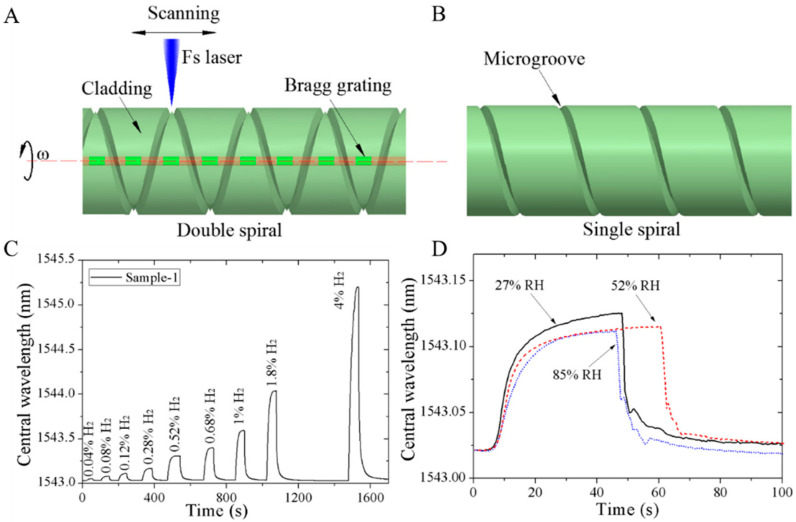
Schematic diagram of fiber spiral microstructure (single spiral and double) hydrogen sensor [39]. (**A**) Laser grooving process and double spiral. (**B**) Single spiral microstructure. (**C**) Hydrogen response (change in central reflection wavelength) under different hydrogen concentrations. (**D**) Hydrogen response with different moisture content. Note: The blue in (**A**) is the pulse laser, the red is the fiber core, and the dotted line is the fiber centerline.

**Figure 12 sensors-24-03146-f012:**
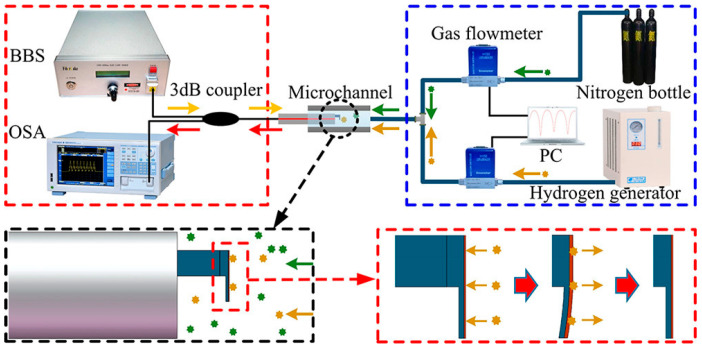
Experimental setup of a cantilever hydrogen sensor [43]. The lower charts show the expansion of the Pd film, causing the test cantilever to bend and return to its original position. Note: The orange-red arrows in the upper part are signals, the brown-green arrows are gases; the green and brown dots and arrows in the lower part are gases and direction, and the red arrows are changes in the sensor head.

**Figure 13 sensors-24-03146-f013:**
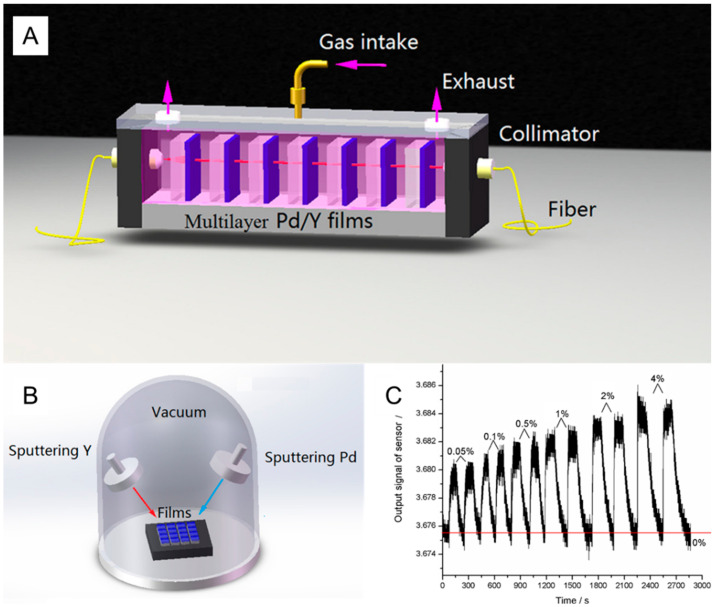
(**A**) Schematic of the multilayer thin-film hydrogen measurement system, (**B**) Diagram illustrating the preparation principle of the hydrogen-sensitive thin film. (**C**) Hydrogen response curve (multiple cycles) [44]. Note: The arrow in (**B**) indicates the direction of sputtering.

**Figure 14 sensors-24-03146-f014:**
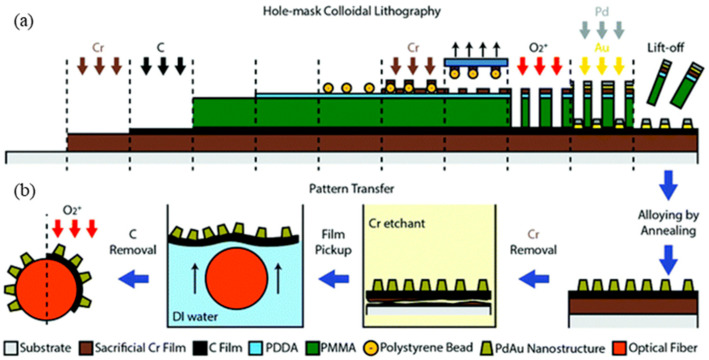
Schematic of the fiber-optic hydrogen sensor preparation using Pd–Au as the hydrogen-sensitive material [48]. (**a**) Sequential growth of the Cr and C layers, followed by the nanofabrication of Pd–Au alloy nanodisk arrays. (**b**) Optical fiber material loading process: begin by removing the organic and Cr layers. Later, allow the nanoparticle-infused C layer to float on the water’s surface. Harnessing the hydrophobic properties of the C layer ensures its adherence to the optical fiber surface. Conclude the process by eliminating the C layer.

**Figure 15 sensors-24-03146-f015:**
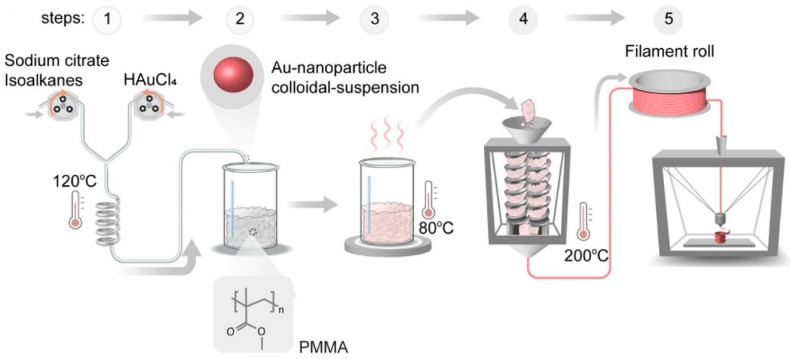
Synthesis process of plasmonic plastic nanocomposites with Au nanospheres [49]. (1) Combine colloidal nanoparticle dispersion with polymer powder. (2) Conduct drying. (3) Proceed with filament extrusion. (4) Melt pressing or 3D printing. (5) FDM 3D printing. Note: Arrows represent processes.

**Figure 16 sensors-24-03146-f016:**
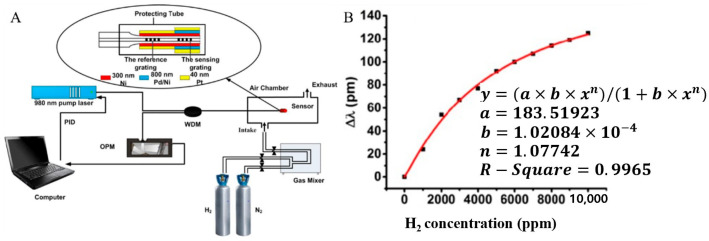
Schematic diagram of the heated FBG hydrogen measuring system (set reference FBG) [45]. (**A**) Optical fiber hydrogen measuring system. (**B**) Wavelength variation curve under different hydrogen concentrations. Note: The image has been slightly formatted.

**Figure 17 sensors-24-03146-f017:**
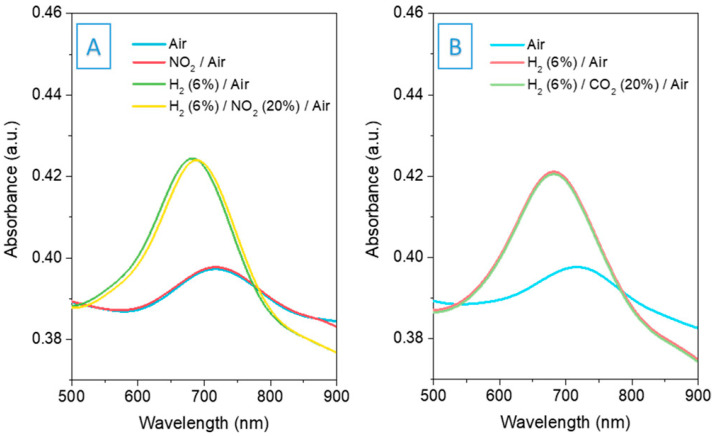
(**A**) Monitoring of 6% hydrogen amidst 20% NO_2_. (**B**) Measurement of 6% hydrogen with 20% CO_2_ present [51].

**Figure 18 sensors-24-03146-f018:**
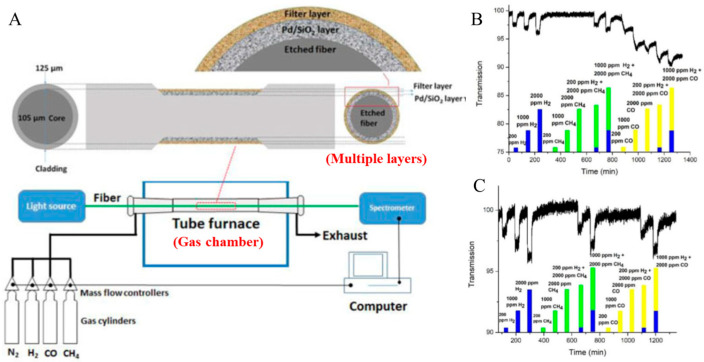
Zeolite filter and Pd-based sensing film optic hydrogen sensors [52]. (**A**) Schematic diagram of the hydrogen gas sensor. (**B**) Response of the hydrogen gas sensor to single gases and gas mixtures (without zeolite membrane). (**C**) Response of the hydrogen gas sensor to single gases and gas mixtures (with zeolite membrane). Note: The curves in (**B**,**C**) represent the transmission, and the histogram represents the gas concentration. (**B**,**C**) show that only in the presence of hydrogen, the transmission will change significantly.

**Figure 19 sensors-24-03146-f019:**
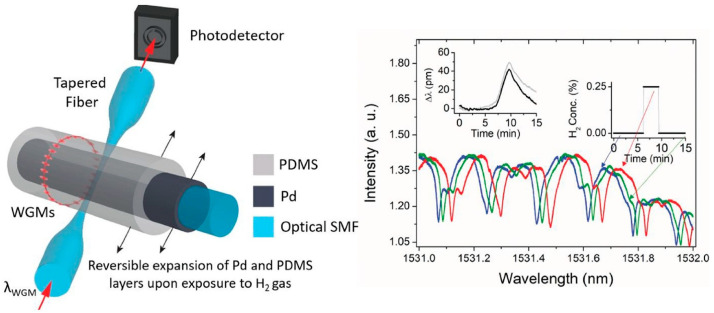
A hydrogen sensor featuring Pd (Pd) as its hydrogen-sensitive material is encompassed by a PDMS ring [58]. The arrows mean light direction.

**Figure 20 sensors-24-03146-f020:**
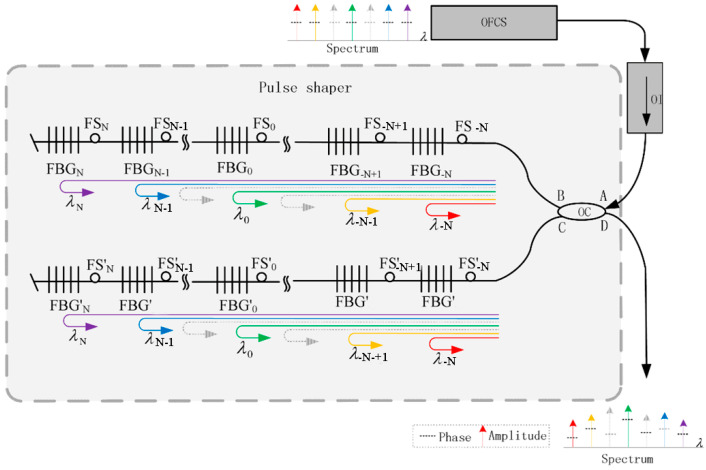
Schematic diagram of an FBG array [60]. Note: The image has been slightly formatted and the arrows mean light direction. The image has been slightly formatted.

**Table 1 sensors-24-03146-t001:** Comparison of fiber hydrogen sensor.

Type	Sensitive Material	Concentration Range	Response Time	Reference
FBG	Pt–WO_3_	0.04–4%	10~20 s	[39]
FBG	Pd	1–5%	0.4 h	[46]
FBG	Pd/Ag	1–4%	300 s	[40]
FBG	(K/S/C)-Pt/WO_3_	0.04–1.5%	50 s	[14]
FBG	Pd	0.5–10%	10 s	[55]
Evanescent	SO_2_	0.02–10%	10 s	[8]
Evanescent	Pd/Au	0.8–4.6%	4.5 s	[56]
Evanescent	MoO_3_	0.125–2%	220 s	[57]
SPR	Pd/SiO_2_/Au	0.5–4%	15 s	[54]
SPR	Pd	0.08–4%	116 s	[36]
MZI	Pd–Au alloy	0.5–20%	200 s	[23]
MZI	Pt–WO_3_	0.1–0.8%	35 s	[15]
FPI	Pt–WO_3_/SO_2_	0.3–2.4%	50 s	[25]
FPI	Pt–WO_3_/SO_2_	0.2–0.8%	23 s	[26]
Micromirror	WO_3_–Pd_2_Pt–Pt	0.01–0.5%	20 s	[31]
Micromirror	WO_3_–Pd_2_Pt–Pt	0.000228–0.2%	200 s	[10]

## Data Availability

No new data were created or analyzed in this study.

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
