# Peer review of "Spectroscopic Techniques and Hydrogen-Sensitive Compounds: A New Horizon in Hydrogen Detection"

_sensors, 2024, doi:10.3390/s24103146_

Round 1

Reviewer 1 Report

Comments and Suggestions for Authors

Dear authors,

your manuscript "Spectroscopic Techniques and Hydrogen-sensitive Compounds: A New Horizon in Hydrogen Detection" represents a comprehensive summary of hydrogen-sensitive materials for optical H2 detection purposes.

The methodology is clear and clearly the paper should be accepted for publication. Before that I'd like to recommend so changes (and minor corrections).

- some references are messed up (reference not found e.g. on line 596)

- in Table 1 it is recommended to give concentration ranges in percent or ppm for quicker comparison

Thanks again for your work

Reviewer 2 Report

Comments and Suggestions for Authors

This review paper presents a comprehensive discussion on hydrogen-sensitive materials and their applications in fiber optic hydrogen sensors. Hydrogen sensitive materials such as metallic oxide (metal attached) and metal, as well as the Mach-Zehnder interferometer, Fabry-Perot interferometer and other fiber structures, and the manufacturing process of hydrogen sensors based on hydrogen sensitive materials are introduced successively. Furthermore, some methods to improve the ability of the sensor to resist environmental interference are introduced. Lastly, it offers prospects for the future development of these sensors from various perspectives. This review paper has a complete structure and clear logic, I think it could be published but it needs minor revision. The following are some issues about this paper:

1. The author needs to be rigorous in the reference, such as the third reference is unrelated to the corresponding sentence, and a space should be left between references and body text:

“in contrast to the direct detection of methane using its absorption spectrum[3]”

In addition, the citation for reference should be placed at the end of the sentence, a space needs to be left between the number and the unit symbol. For example:

“Yan [8] reported nanocrystalline Tin oxide thin films for hydrogen sensing based on an optical fiber evanescent field. Pt-WO3 is particularly sensitive in gas detection, with a sensitivity of over 1nm of 1% H2 increased [9] and an excellent response time of less than 1 minute in the experiment.”

2. Authors need to carefully check the related problems:

In the title of Fig 1, only the first letter of the first word needs to be capitalized.

In Fig 3, it is more appropriate to replace “coupler” with “circulator”.

In Fig 3, 4, 7, 9 and 10, a period is missing at the end of the title.

Line 83: “WO3” needs to be modified to “WO3”.

Line 196-197: “seconds” should be “s”.

Line 222-224: “The conventional dual fiber (MZI) hydrogen sensors can be further improved by enhancing the material properties used for gas testing.” Please confirm this sentence.

In Fig 14, it’s title should be be left justified.

Line 596, 628-629: “Error! Reference source not found.” Please confirm this sentence.

Line 653, 673: “Conclusion” should be “7.1 Conclusion”; “7. Outlook” should be “7.1 Outlook”.

Table 1: “FRI” should be “FPI”.

Except for the first word “palladium”, the other “palladium” is replaced by its abbreviation “Pd”.

3. When talking about the work done in some papers, authors should try to indicate the specific number of response sensitivity to hydrogen, response time and other parameters. For example, in lines 137-143, the author only praised the three-layer structure, however, the specific number can give the reader a more intuitive feeling.

4. This paper focused on the Spectroscopic Techniques for H2 sensing, but unfortunately the authors missed some important publications for H2 detection based on laser spectroscopy. They should be added. 1) Hydrogen sensor based on tunable diode laser absorption spectroscopy; 2) Highly sensitive and fast hydrogen detection based on light-induced thermoelastic spectroscopy.

Reviewer 3 Report

Comments and Suggestions for Authors

A review of optical hydrogen sensors is very useful. However, i have some comments about your manuscript:

Line 10: other hydrogen sensors are not sensitive to magnetic fields, so this is not an advantage of hydrogen sensors.

Line 32: metal oxide and electrochemical sensors do not generate or require voltages above a few volts so there is no electrode sparks. Additionally, optical sensors using lasers must now be proven safe: the allowed optical power is defined in EN 60079-28. Optical sensors are not intrinsically safe.

Line 81 and elsewhere: they are metal oxides, not metallic oxides.

Line 120: the sensing material is palladium, not graphene, which is used only as a supporting substrate. Palladium is the most important sensing material for hydrogen detection due to its ability to reversibly alloy with protons.

Line 149: please define what you mean by "advanced hydrogen measurement system".

Line 165: Why are UV etched FBGs not suitable for high temperature applications?

Line 180: Pd is a reactant, not a catalyst.

Line 196: please correct £1.597 dB%".

Line 253: Who is referred to: "I developed..."

Lines 258 and 308: 0%-2.4%: what is the LOD? Also, you refer to both LDL and LOD in this text.

Line 307: "shift of 290 pm" referenced to what?

Line 452: "-2nm%-1"   please rephrase

Line 494: 0.5nm: referenced to what?

Lines 596, 628: references missing

Table 1: the critical parameter is the LOD, but some of the entries do not list LOD.

Lines 636-639: discussion of high concentration: these two sentences contradict each other.

Line 680: making dense layers may improve mechanical strength but it also frequently increases response time to  unacceptable long times.

Conclusion: you refer to graphene yet is is not important in hydrogen sensors. You do not emphasise the importance of palladium, you do not include in the conclusion the importance of temperature control in these optical sensors.

Polymer section: this section is incorrect. you state that "polymers improve the range". How? Polymers are not intrinsically a sensing material for hydrogen, they are substrates and mechanical support.

Comments on the Quality of English Language

There are many phrases in this manuscript that are not used in scientific papers: nuanced grasp; well rounded perspective; paramount importance; nuanced deformations (optical material property change); discernable (can be measured);anchored in the optical spectrum (using the optical spectrum); a deeply facilitating expansive measurement coverage; indifferent to humidity (insensitive to humidity)

Line 90-: what does "suitable security proof" mean?

Lines 47 to 65: please rewrite this section. 

Lines 99-108: repetitive sentences. Does not explain the difference between WO3 and other metal oxides and Pd.

Line 157: Should be "monitoring central wavelength"

Line 162: please correct "braidings"

Line 398: to correct Pd (d is not subscript)

Line 471: remove precisely

Heading 3.8: Sensor Structure, not Sensor's Structure

Line 475 and 507: Lower Detection Limit (not low or lowest)

Line 488 and 493: sputter, not spatter

Line 516: change a to the

Line523: "produce sensors for response" please rewrite.

Line 686: Change "Polymer is" to "Polymers are"

Section 5.1 includes many AI generated verbs and adverbs. Please rewrite.

Round 2

Reviewer 3 Report

Comments and Suggestions for Authors

Thank you for replying so thoroughly, the manuscript is improved.

My only comment is your first change that resistive sensors (electrochemical sensors are amperometric, not resistive, so I assume you are referring to metal oxides) have a shorter detection range and higher usage costs is incorrect. The Beer-Lambert law generally limits transmissive optical gas sensors to about one decade of concentration, while chemiresistors have up to 3 decades of resistive changes corresponding to more than two decades of concentrations. Cost of metal oxides is typically $3 to $20 while optical sensors are much more expensive and metal oxides are used in cars because they have a long lifetime.

Comments on the Quality of English Language

Improved. I assume the journal editor will review again.
